# Posterior Matching for Arbitrary Conditioning

**Ryan R. Strauss**
Department of Computer Science
UNC at Chapel Hill
rrs@cs.unc.edu

**Junier B. Oliva**
Department of Computer Science
UNC at Chapel Hill
joliva@cs.unc.edu

## Abstract

Arbitrary conditioning is an important problem in unsupervised learning, where we seek to model the conditional densities $p(\mathbf{x}_u \mid \mathbf{x}_o)$ that underly some data, for all possible non-intersecting subsets $o, u \subset \{1, \ldots, d\}$. However, the vast majority of density estimation only focuses on modeling the joint distribution $p(\mathbf{x})$, in which important conditional dependencies between features are opaque. We propose a simple and general framework, coined Posterior Matching, that enables Variational Autoencoders (VAEs) to perform arbitrary conditioning, without modification to the VAE itself. Posterior Matching applies to the numerous existing VAE-based approaches to joint density estimation, thereby circumventing the specialized models required by previous approaches to arbitrary conditioning. We find that Posterior Matching is comparable or superior to current state-of-the-art methods for a variety of tasks with an assortment of VAEs (e.g. discrete, hierarchical, VaDE).

## 1 Introduction

Variational Autoencoders (VAEs) [21] are a widely adopted class of generative model that have been successfully employed in numerous areas [4, 15, 26, 33, 16]. Much of their appeal stems from their ability to probabilistically represent complex data in terms of lower-dimensional latent codes.

Like most other generative models, VAEs are typically designed to model the joint data distribution, which communicates likelihoods for particular configurations of all features at once. This can be useful for some tasks, such as generating images, but the joint distribution is limited by its inability to explicitly convey the conditional dependencies between features. In many cases, conditional distributions, which provide the likelihood of an event given some known information, are more relevant and useful. Conditionals can be obtained in theory by marginalizing the joint distribution, but in practice, this is generally not analytically available and is expensive to approximate.

Easily assessing the conditional distribution over *any subset* of features is important for tasks where decisions and predictions must be made over a varied set of possible information. For example, some medical applications may require reasoning over: the distribution of *blood pressure* given *age* and *weight*; or the distribution of *heart-rate* and *blood-oxygen level* given *age*, *blood pressure*, and *BMI*; etc. For flexibility and scalability, it is desirable for a *single* model to provide all such conditionals at inference time. More formally, this task is known as *arbitrary conditioning*, where the goal is to model the conditional density $p(\mathbf{x}_u \mid \mathbf{x}_o)$ for any arbitrary subsets of unobserved features $\mathbf{x}_u$ and observed features $\mathbf{x}_o$. In this work, we show, by way of a simple and general framework, that traditional VAEs can perform arbitrary conditioning, without modification to the VAE model itself.

Our approach, which we call Posterior Matching, is to model the distribution $p(\mathbf{z} \mid \mathbf{x}_o)$ that is induced by some VAE, where $\mathbf{z}$ is the latent code. In other words, we consider the distribution of latent codes given partially observed features. We do this by having a neural network output an approximate partially observed posterior $q(\mathbf{z} \mid \mathbf{x}_o)$. In order to train this network, we develop a straightforward maximum likelihood estimation objective and show that it is equivalent to maximizing $p(\mathbf{x}_u \mid \mathbf{x}_o)$,

36th Conference on Neural Information Processing Systems (NeurIPS 2022).

the quantity of interest. Unlike prior works that use VAEs for arbitrary conditioning, we do not make special assumptions or optimize custom variational lower bounds. *Rather, training via Posterior Matching is simple, highly flexible, and without limiting assumptions on approximate posteriors (e.g., $q(\mathbf{z} \mid \mathbf{x}_o)$ need not be reparameterized and can thus be highly expressive).*

We conduct several experiments in which we apply Posterior Matching to various types of VAEs for a myriad of different tasks, including image inpainting, tabular arbitrary conditional density estimation, partially observed clustering, and active feature acquisition. We find that Posterior Matching leads to improvements over prior VAE-based methods across the range of tasks we consider.

## 2   Background

**Arbitrary Conditioning**  A core problem in unsupervised learning is *density estimation*, where we are given a dataset $\mathcal{D} = \{\mathbf{x}^{(i)}\}_{i=1}^{N}$ of *i.i.d.* samples drawn from an unknown distribution $p(\mathbf{x})$ and wish to learn a model that best approximates the probability density function $p$. A limitation of only learning the joint distribution $p(\mathbf{x})$ is that it does not provide direct access to the conditional dependencies between features. *Arbitrary conditional density estimation* [18, 24, 38] is a more general task where we want to estimate the conditional density $p(\mathbf{x}_u \mid \mathbf{x}_o)$ for all possible subsets of observed features $o \subset \{1, \dots, d\}$ and unobserved features $u \subset \{1, \dots, d\}$ such that $o$ and $u$ do not intersect. Here, $\mathbf{x}_o \in \mathbb{R}^{|o|}$ and $\mathbf{x}_u \in \mathbb{R}^{|u|}$. Estimation of joint or marginal likelihoods is a special case where $o = \emptyset$. Note that, while not strictly necessary for arbitrary conditioning methods [24, 38], we assume $\mathcal{D}$ is fully observed, a requirement for training traditional VAEs.

**Variational Autoencoders**  Variational Autoencoders (VAEs) [21] are a class of generative models that assume a generative process in which data likelihoods are represented as $p(\mathbf{x}) = \int p(\mathbf{x} \mid \mathbf{z})p(\mathbf{z})\,\mathrm{d}\mathbf{z}$, where $\mathbf{z}$ is a latent variable that typically has lower dimensionality than the data $\mathbf{x}$. A tractable distribution that affords easy sampling and likelihood evaluation, such as a standard Gaussian, is usually imposed on the prior $p(\mathbf{z})$. These models are learned by maximizing the *evidence lower bound* (ELBO) of the data likelihood:

$$\log p(\mathbf{x}) \geq \mathbb{E}_{\mathbf{z} \sim q_\psi(\cdot \mid \mathbf{x})}[\log p_\phi(\mathbf{x} \mid \mathbf{z})] - \mathrm{KL}(q_\psi(\mathbf{z} \mid \mathbf{x}) \,||\, p(\mathbf{z})),$$

where $q_\psi(\mathbf{z} \mid \mathbf{x})$ and $p_\phi(\mathbf{x} \mid \mathbf{z})$ are the encoder (or approximate posterior) and decoder of the VAE, respectively. The encoder and decoder are generally neural networks that output tractable distributions (e.g., a multivariate Gaussian). In order to properly optimize the ELBO, samples drawn from $q_\psi(\mathbf{z} \mid \mathbf{x})$ must be differentiable with respect to the parameters of the encoder (often called the *reparameterization trick*). After training, a new data point $\hat{\mathbf{x}}$ can be easily generated by first sampling $\mathbf{z}$ from the prior, then sampling $\hat{\mathbf{x}} \sim p_\phi(\cdot \mid \mathbf{z})$.

## 3   Posterior Matching

In this section we describe our framework, coined Posterior Matching, to model the underlying arbitrary conditionals in a VAE. In many respects, Posterior Matching *cuts the Gordian knot* to uncover the conditional dependencies. Following our insights, we show that our approach is *direct* and *intuitive*. Notwithstanding, we are the first to apply this direct methodology for arbitrary conditionals in VAEs and are the first to connect our proposed loss with arbitrary conditional likelihoods $p(\mathbf{x}_u \mid \mathbf{x}_o)$. Note that we are *not* proposing a new type of VAE. Rather, we are formalizing a simple and intuitive methodology that can be applied to numerous existing (or future) VAEs.

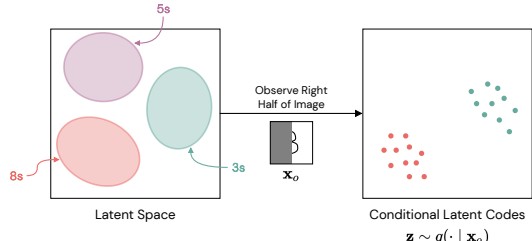

Figure 1: A two-dimensional VAE latent space (left) that represents the distribution of codes of handwritten 3s, 5s, and 8s. Conditioning on the subset of pixels shown in the center should then result in a distribution over latent codes that only correspond to 3s and 8s, as shown on the right.

## 3.1 Motivation

Let us begin with a motivating example, depicted in Figure 1. Suppose we have trained a VAE on images of handwritten 3s, 5s, and 8s. This VAE has thus learned to represent these images in a low-dimensional latent space. Any given code (vector) in this latent space represents a distribution over images in the original data space, which can be retrieved by passing that code through the VAE's decoder. Some regions in the latent space will contain codes that represent 3s, some will represent 5s, and some will represent 8s. There is typically only an interest in mapping from a given image $\mathbf{x}$ to a distribution over the latent codes that could represent that image, i.e., the posterior $q(\mathbf{z} \mid \mathbf{x})$. *However, we can just as easily ask which latent codes are feasible having only observed part of an image.*

For example, if we only see the right half the image shown in Figure 1, we know the digit could be a 3 or an 8, but certainly not a 5. Thus, the distribution over latent codes that could correspond to the full image, that is $p_\psi(\mathbf{z} \mid \mathbf{x}_o)$ (where $\psi$ is the encoder's parameters), should only include regions that represent 3s or 8s. Decoding any sample from $p_\psi(\mathbf{z} \mid \mathbf{x}_o)$ will produce an image of a 3 or an 8 that aligns with what has been observed.

The important *insight* is that we can think about how conditioning on $\mathbf{x}_o$ changes the distribution over latent codes without explicitly worrying about what the (potentially higher-dimensional and more complicated) conditional distribution over $\mathbf{x}_u$ looks like. Once we know $p_\psi(\mathbf{z} \mid \mathbf{x}_o)$, we can easily move back to the original data space using the decoder.

## 3.2 Approximating the Partially Observed Posterior

The partially observed approximate posterior of interest is not readily available, as it is implicitly defined by the VAE:

$$p_\psi(\mathbf{z} \mid \mathbf{x}_o) = \mathbb{E}_{\mathbf{x}_u \sim p(\cdot \mid \mathbf{x}_o)} \Big[ q_\psi(\mathbf{z} \mid \mathbf{x}_o, \mathbf{x}_u) \Big], \tag{1}$$

where $q_\psi(\mathbf{z} \mid \mathbf{x}_o, \mathbf{x}_u) = q_\psi(\mathbf{z} \mid \mathbf{x})$ is the VAE's encoder. Thus, we introduce a neural network in order to approximate it.

Given a network that outputs the distribution $q_\theta(\mathbf{z} \mid \mathbf{x}_o)$ (i.e. the partially observed encoder in Figure 2), we now discuss our approach to training it. Our approach is guided by the priorities of simplicity and generality. We minimize (with respect to $\theta$) the following likelihoods, where the samples are coming from our target distribution as defined in Equation 1:

$$\mathbb{E}_{\mathbf{x}_u \sim p(\cdot \mid \mathbf{x}_o)} \Big[ \mathbb{E}_{\mathbf{z} \sim q_\psi(\cdot \mid \mathbf{x}_o, \mathbf{x}_u)} [- \log q_\theta(\mathbf{z} \mid \mathbf{x}_o)] \Big]. \tag{2}$$

We discuss how this is optimized in practice in Section 3.4.

Due to the relationship between negative log-likelihood minimization and KL-divergence minimization [3], we can interpret Equation 2 as minimizing:

$$\mathbb{E}_{\mathbf{x}_u \sim p(\cdot \mid \mathbf{x}_o)} \Big[ \mathrm{KL} \left( q_\psi(\mathbf{z} \mid \mathbf{x}_o, \mathbf{x}_u) \mid\mid q_\theta(\mathbf{z} \mid \mathbf{x}_o) \right) \Big]. \tag{3}$$

We can directly minimize the KL-divergence in Equation 3 if it is analytically available between the two posteriors, for instance if both posteriors are Gaussians. However, Equation 2 is more general in that it allows us to use more expressive (e.g., autoregressive) distributions for $q_\theta(\mathbf{z} \mid \mathbf{x}_o)$ with which the KL-divergence cannot be directly computed. This is important given that $p_\psi(\mathbf{z} \mid \mathbf{x}_o)$ is likely to be complex (e.g., multimodal) and not easily captured by a Gaussian (as in Figure 1). Importantly, there is no requirement for $q_\theta(\mathbf{z} \mid \mathbf{x}_o)$ to be reparameterized, which would further limit the class of distributions that can be used. There is a high degree of flexibility in the choice of distribution for the partially observed posterior. Note that this objective does not utilize the decoder.

## 3.3 Connection with Arbitrary Conditioning

While the Posterior Matching objective from Equation 2 and Equation 3 is intuitive, it is not immediately clear how this approach relates back to the arbitrary conditioning objective of maximizing $p(\mathbf{x}_u \mid \mathbf{x}_o)$. We formalize this connection in Theorem 3.1 (see Appendix for proof).

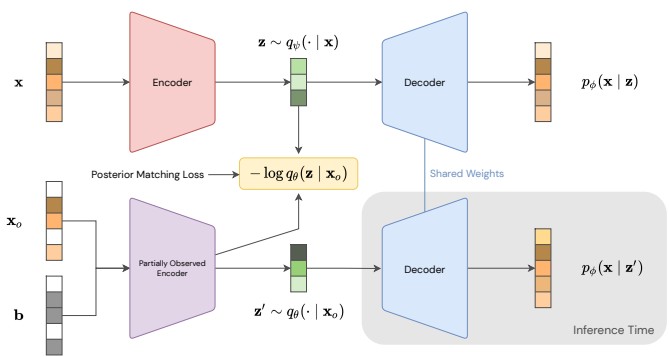

Figure 2: Overview of the Posterior Matching framework. The partially observed encoder can be appended to any existing VAE to enable arbitrary conditioning. At inference time, we can decode samples from $q_\theta(\mathbf{z} \mid \mathbf{x}_o)$ to perform imputation or compute $p(\mathbf{x}_u \mid \mathbf{x}_o)$.

**Theorem 3.1.** *Let $q_\psi(\mathbf{z} \mid \mathbf{x})$ and $p_\phi(\mathbf{x} \mid \mathbf{z})$ be the encoder and decoder, respectively, for some VAE. Additionally, let $q_\theta(\mathbf{z} \mid \mathbf{x}_o)$ be an approximate partially observed posterior. Then minimizing $\mathbb{E}_{\mathbf{x}_u \sim p(\cdot \mid \mathbf{x}_o)} \Big[ KL\big( q_\psi(\mathbf{z} \mid \mathbf{x}_o, \mathbf{x}_u) \,\|\, q_\theta(\mathbf{z} \mid \mathbf{x}_o) \big) \Big]$ is equivalent to minimizing*

$$\mathbb{E}_{\mathbf{x}_u \sim p(\cdot \mid \mathbf{x}_o)} \Big[ -\log p_{\theta,\phi}(\mathbf{x}_u \mid \mathbf{x}_o) + KL\big( q_\psi(\mathbf{z} \mid \mathbf{x}_o, \mathbf{x}_u) \,\|\, q_\theta(\mathbf{z} \mid \mathbf{x}_o, \mathbf{x}_u) \big) \Big], \qquad (4)$$

*with respect to the parameters $\theta$.*

The first term inside the expectation in Equation 4 gives us the explicit connection back to the arbitrary conditioning likelihood $p(\mathbf{x}_u \mid \mathbf{x}_o)$, which is being maximized when minimizing Equation 4. The second term acts as a sort of regularizer by trying to make the partially observed posterior match the VAE posterior when conditioned on all of $\mathbf{x}$ — intuitively, this makes sense as a desirable outcome.

### 3.4 Implementation

A practical training loss follows quickly from Equation 2. For the outer expectation, we do not have access to the true distribution $p(\mathbf{x}_u \mid \mathbf{x}_o)$, but for a given instance $\mathbf{x}$ that has been partitioned into $\mathbf{x}_o$ and $\mathbf{x}_u$, we do have one sample from this distribution, namely $\mathbf{x}_u$. So we approximate this expectation using $\mathbf{x}_u$ as a single sample. This type of single-sample approximation is common with VAEs, e.g., when estimating the ELBO. For the inner expectation, we have access to $q_\psi(\mathbf{z} \mid \mathbf{x})$, which can easily be sampled in order to estimate the expectation. In practice, we generally use a single sample for this as well. This gives us the following Posterior Matching loss:

$$\mathcal{L}_{\mathrm{PM}}(\mathbf{x}, o, \theta, \psi) = -\mathbb{E}_{\mathbf{z} \sim q_\psi(\cdot \mid \mathbf{x})} \big[ \log q_\theta(\mathbf{z} \mid \mathbf{x}_o) \big], \qquad (5)$$

where $o$ is the set of observed feature indices. During training, $o$ can be randomly sampled from a problem-specific distribution for each minibatch.

Figure 2 provides a visual overview of our approach. In practice, we represent $\mathbf{x}_o$ as a concatenation of $\mathbf{x}$ that has had unobserved features set to zero and a bitmask $\mathbf{b}$ that indicates which features are observed. This representation has been successful in other arbitrary conditioning models [24, 38]. However, this choice is not particularly important to Posterior Matching itself, and alternative representations, such as set embeddings, are valid as well.

As required by VAEs, samples from $q_\psi(\mathbf{z} \mid \mathbf{x})$ will be reparameterized, which means that minimizing $\mathcal{L}_{\mathrm{PM}}$ will influence the parameters of the VAE's encoder in addition to the partially observed posterior network. In some cases, this may be advantageous, as the encoder can be guided towards learning a latent representation that is more conducive to arbitrary conditioning. However, it might also be desirable to train the VAE independently of the partially observed posterior, in which case we can choose to stop gradients on the samples $\mathbf{z} \sim q_\psi(\cdot \mid \mathbf{x})$ when computing $\mathcal{L}_{\mathrm{PM}}$.

Similarly, the partially observed posterior can be trained against an existing pretrained VAE. In this case, the parameters of the VAE's encoder and decoder are frozen, and we only optimize $\mathcal{L}_{\mathrm{PM}}$ with respect to $\theta$. Otherwise, we jointly optimize the VAE's ELBO and $\mathcal{L}_{\mathrm{PM}}$.

We emphasize that there is a high degree of flexibility with the choice of VAE, i.e. we have not imposed any unusual constraints. However, there are some potentially limiting practical considerations that have not been explicitly mentioned yet. First, the training data must be fully observed, as with traditional VAEs, since $\mathcal{L}_{\text{PM}}$ requires sampling $q_\psi(\mathbf{z} \mid \mathbf{x})$. However, given that the base VAE requires fully observed training data anyway, this is generally not a relevant limitation for our purposes. Second, it is *convenient* in practice for the VAE's decoder to be factorized, i.e. $p(\mathbf{x} \mid \mathbf{z}) = \prod_i p(x_i \mid \mathbf{z})$, as this allows us to easily sample from $p(\mathbf{x}_u \mid \mathbf{z})$ (sampling $\mathbf{x}_u$ is less straightforward with other types of decoders). However, it is standard practice to use factorized decoders with VAEs, so this is ordinarily not a concern. We also note that, while useful for easy sampling, a factorized decoder is *not necessary* for optimizing the Posterior Matching loss, which does not incorporate the decoder.

### 3.5 Posterior Matching Beyond Arbitrary Conditioning

The concept of matching VAE posteriors is quite general and has other uses beyond the application of arbitrary conditioning. We consider one such example, which still has ties to arbitrary conditioning, in order to give a flavor for other potential uses.

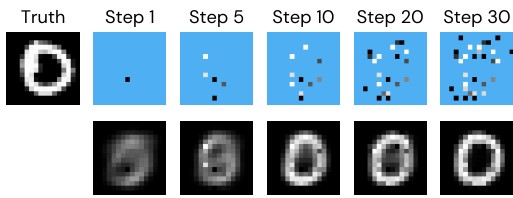

A common application of arbitrary conditioning is *active feature acquisition* [13, 23, 25], where informative features are sequentially acquired on an instance-by-instance basis. In the unsupervised case, the aim is to acquire as few features as possible while maximizing the ability to reconstruct the remaining unobserved features (see Figure 3 for example).

Figure 3: Example acquisitions from our CNN model using the lookahead posteriors (see Section 5.5). Top row shows $\mathbf{x}_o$, and blue pixels are unobserved. Bottom row shows imputation of $\mathbf{x}_u$.

One approach to active feature acquisition is to greedily select the feature that will maximize the expected amount of information to be gained about the currently unobserved features [23, 25]. For VAEs, Ma et al. [25] show that this is equivalent to selecting each feature according to

$$\operatorname*{argmax}_{i \in u} H(\mathbf{z} \mid \mathbf{x}_o) - \mathbb{E}_{x_i \sim p(\cdot \mid \mathbf{x}_o)}\Big[H(\mathbf{z} \mid \mathbf{x}_o, x_i)\Big] = \operatorname*{argmin}_{i \in u} \mathbb{E}_{x_i \sim p(\cdot \mid \mathbf{x}_o)}\Big[H(\mathbf{z} \mid \mathbf{x}_o, x_i)\Big]. \quad (6)$$

For certain families of posteriors, such as multivariate Gaussians, the entropies in Equation 6 can be analytically computed. In practice, approximating the expectation in Equation 6 is done via entropies of the posteriors $p^{(i)}(\mathbf{z} \mid \mathbf{x}_o) \equiv \mathbb{E}_{x_i \sim p_{\theta,\phi}(\cdot \mid \mathbf{x}_o)}\Big[q_\theta(\mathbf{z} \mid \mathbf{x}_o, x_i)\Big]$, where samples from $p_{\theta,\phi}(x_i \mid \mathbf{x}_o)$ are produced by first sampling $\mathbf{z} \sim q_\theta(\cdot \mid \mathbf{x}_o)$ and then passing $\mathbf{z}$ through the VAE's decoder $p_\phi(x_i \mid \mathbf{z})$ (we call $p^{(i)}(\mathbf{z} \mid \mathbf{x}_o)$ the "lookahead" posterior for feature $i$, since it is obtained by imagining what the posterior will look like after one acquisition into the future). Hence, computing the resulting entropies requires one network evaluation per sample of $x_i$ to encode $z$, for $i \in u$. Thus, if using $k$ samples for each $x_i$, each greedy step will be $\Omega(k \cdot |u|)$, which may be prohibitive in high dimensions.

In analogous fashion to the Posterior Matching approach that has already been discussed, we can train a neural network to directly output the lookahead posteriors for all features at once. The Posterior Matching loss in this case is

$$\mathcal{L}_{\text{PM-Lookahead}}(\mathbf{x}, o, u, \omega, \theta, \phi) = \sum_{i \in u} \mathbb{E}_{x_i \sim p_{\theta,\phi}(\cdot \mid \mathbf{x}_o)}\bigg[\mathbb{E}_{\mathbf{z} \sim q_\theta(\cdot \mid \mathbf{x}_o, x_i)}\Big[-\log q_\omega^{(i)}(\mathbf{z} \mid \mathbf{x}_o)\Big]\bigg], \quad (7)$$

where $\omega$ is the parameters of the lookahead posterior network. In practice, we train a single shared network with a final output layer that outputs the parameters of all $q_\omega^{(i)}(\mathbf{z} \mid \mathbf{x}_o)$. Note that given the distributions $q_\omega^{(i)}(\mathbf{z} \mid \mathbf{x}_o)$ for all $i$, computing the greedy acquisition choice consists of doing a forward evaluation of our network, then choosing the feature $i \in u$ such that the entropy of $q_\omega^{(i)}(\mathbf{z} \mid \mathbf{x}_o)$ is minimized. In other words, we may bypass the individual samples of $x_i$, and use a single shared network for a faster acquisition step. In this setting, we let $q_\omega^{(i)}(\mathbf{z} \mid \mathbf{x}_o)$ be a multivariate Gaussian so that the entropy computation is trivial. See Appendix for a diagram of the entire process. This use of Posterior Matching leads to large improvements in the computational efficiency of greedy active feature acquisition (demonstrated empirically in Section 5.5).

# 4 Prior Work

A variety of approaches to arbitrary conditioning have been previously proposed. ACE is an autoregressive, energy-based method that is the current state-of-the-art for arbitrary conditional likelihood estimation and imputation, although it can be computationally intensive for very high dimensional data [38]. ACFlow is a variant of normalizing flows that can give analytical arbitrary conditional likelihoods [24]. Several other methods, including Sum-Product Networks [32, 6], Neural Conditioner [2], and Universal Marginalizer [10], also have the ability to estimate conditional likelihoods.

Rezende et al. [34] were among the first to suggest that VAEs can be used for imputation. More recently, VAEAC was proposed as a VAE variant designed for arbitrary conditioning [18]. Unlike Posterior Matching, VAEAC is not a general framework and cannot be used with typical pretrained VAEs. EDDI is a VAE-based approach to active feature acquisition and relies on arbitrary conditioning [25]. The authors introduce a "Partial VAE" in order to perform the arbitrary conditioning, which, similarly to Posterior Matching, tries to model $p(\mathbf{z} \mid \mathbf{x}_o)$. Unlike Posterior Matching, they do this by maximizing a variational lower bound on $p(\mathbf{x}_o)$ using a partial inference network $q(\mathbf{z} \mid \mathbf{x}_o)$ (there is no standard VAE posterior $q(\mathbf{z} \mid \mathbf{x})$ in EDDI). Gong et al. [13] use a similar approach that is based on the Partial VAE of EDDI. The major drawback of these methods is that, unlike with Posterior Matching, $q(\mathbf{z} \mid \mathbf{x}_o)$ must be reparameterizable in order to optimize the lower bound (the authors use a diagonal Gaussian). Thus, certain more expressive distributions (e.g., autoregressive) cannot be used. Additionally, these methods cannot be applied to existing VAEs. The methods of Ipsen et al. [17] and Collier et al. [9] are also similar to EDDI, where the former optimizes an approximation of $p(\mathbf{x}_o, \mathbf{b})$ and the latter optimizes a lower bound on $p(\mathbf{x}_o \mid \mathbf{b})$. Ipsen et al. [17] also focuses on imputation for data that is missing "not at random", a setting that is outside the focus of our work.

There are also several works that have considered learning to identify desirable regions in latent spaces. Engel et al. [11] start from a pretrained VAE, but then train a separate GAN [14] with special regularizers to do their conditioning. They only condition on binary vectors, $\mathbf{y}$, that correspond to a small number of predefined attributes, whereas we allow for conditioning on arbitrary subsets of continuous features $\mathbf{x}_o$ (a more complicated conditioning space). Also, their resulting GAN does not make the likelihood $q(\mathbf{z} \mid \mathbf{y})$ available, whereas Posterior Matching directly (and flexibly) models $q(\mathbf{z} \mid \mathbf{x}_o)$, which may be useful for downstream tasks (e.g. Section 5.5) and likelihood evaluation (see Appendix). Furthermore, Posterior Matching trains directly through KL, without requiring an additional critic. Whang et al. [40] learn conditional distributions, but not *arbitrary* conditional distributions (a much harder problem). They also consider normalizing flow models, which are limited to invertible architectures with tractable Jacobian determinants and latent spaces that have the same dimensionality as the data (unlike VAEs). Cannella et al. [7] similarly do conditional sampling from a model of the joint distribution, but are also restricted to normalizing flow architectures and require a more expensive MCMC procedure for sampling.

# 5 Experiments

In order to empirically test Posterior Matching, we apply it to a variety of VAEs aimed at different tasks. We find that our models are able to match or surpass the performance of previous specialized VAE methods. All experiments were conducted using JAX [5] and the DeepMind JAX Ecosystem [1]. Code is available at `https://github.com/lupalab/posterior-matching`.

Our results are dependent on the choice of VAE, and the particular VAEs used in our experiments were *not* the product of extensive comparisons and did not undergo thorough hyperparameter tuning — that is not the focus of this work. With more carefully selected or tuned VAEs, and as new VAEs continue to be developed, we can expect Posterior Matching's downstream performance to improve accordingly on any given task. We emphasize that our experiments span a *diverse set of task, domains, and types of VAE*, wherein Posterior Matching was effective.

## 5.1 MNIST

In this first experiment, our goal is to demonstrate that Posterior Matching replicates the intuition depicted in Figure 1. We do this by training a convolutional VAE with Posterior Matching on the MNIST dataset. The latent space of this VAE is then mapped to two dimensions with UMAP [27] and visualized in Figure 4. In the figure, black points represent samples from $q_\theta(\mathbf{z} \mid \mathbf{x}_o)$, and for select

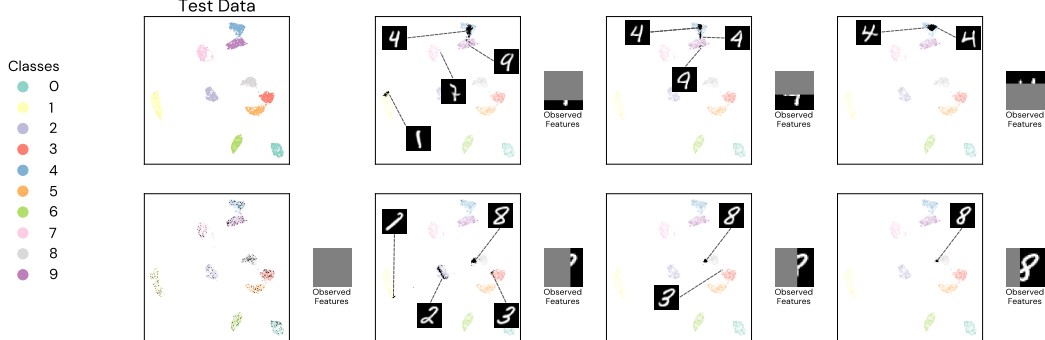

Figure 4: UMAP [27] visualization of the latent space of a VAE trained on MNIST. Black dots represent samples from the distribution $q_\theta(\mathbf{z} \mid \mathbf{x}_o)$ learned via Posterior Matching. Images of $\mathbf{x}_o$ are to the right of their respective plot. Samples from each mode are decoded and shown.

Table 1: Peak signal-to-noise ratio (PSNR) and precision/recall scores [35] for image inpaintings. We report mean and standard deviation across five evaluations with different random masks. VAEAC and ACFlow results are taken from Li et al. [24]. Higher is better for all metrics.

| | MNIST | | | OMNIGLOT | | | CELEBA | | |
|---|---|---|---|---|---|---|---|---|---|
| | PSNR | Precision | Recall | PSNR | Precision | Recall | PSNR | Precision | Recall |
| VDVAE + PM (ours) | **21.603 ± 0.022** | **0.996** | **0.996** | 18.256 ± 0.038 | **0.995** | **0.994** | **27.190 ± 0.049** | **0.995** | **0.995** |
| VQ-VAE + PM (ours) | 19.981 ± 0.021 | 0.989 | 0.993 | 17.954 ± 0.046 | 0.979 | 0.973 | 25.531 ± 0.036 | 0.982 | 0.984 |
| VAEAC | 19.613 ± 0.042 | 0.877 | 0.975 | 17.693 ± 0.023 | 0.525 | 0.926 | 23.656 ± 0.027 | 0.966 | 0.967 |
| ACFlow | 17.349 ± 0.018 | 0.945 | 0.984 | 15.572 ± 0.031 | 0.962 | 0.971 | 22.393 ± 0.040 | 0.970 | 0.988 |
| ACFlow+BG | 20.828 ± 0.031 | 0.947 | 0.983 | **18.838 ± 0.009** | 0.967 | 0.970 | 25.723 ± 0.020 | 0.964 | 0.987 |

samples, the corresponding reconstruction is shown. The encoded test data is shown, colored by true class label, to highlight which regions correspond to which digits. We see that the experimental results nicely replicate our earlier intuitions — the learned distribution $q_\theta(\mathbf{z} \mid \mathbf{x}_o)$ puts probability mass only in parts of the latent space that correspond to plausible digits based on what is observed and successfully captures multimodal distributions (see the second column in Figure 4).

## 5.2 Image Inpainting

One practical application of arbitrary conditioning is image inpainting, where only part of an image is observed and we want to fill in the missing pixels with visually coherent imputations. As with prior works [18, 24], we assume pixels are missing completely at random. We test Posterior Matching as an approach to this task by pairing it with both discrete and hierarchical VAEs.

**Vector Quantized-VAEs** We first consider VQ-VAE [30], a type of VAE that is known to work well with images. VQ-VAE differs from the typical VAE with its use of a discrete latent space. That is, each latent code is a grid of discrete indices rather than a vector of continuous values. Because the latent space is discrete, Oord et al. [30] model the prior distribution with a PixelCNN [29, 36] after training the VQ-VAE. We similarly use a conditional PixelCNN to model $q_\theta(\mathbf{z} \mid \mathbf{x}_o)$. First, a convolutional network maps $\mathbf{x}_o$ to a vector, and that vector is then used as a conditioning input to the PixelCNN. More architecture and training details can be found in the Appendix. We train VQ-VAEs with Posterior Matching for the MNIST, OMNIGLOT, and CELEBA datasets. Table 1 reports peak signal-to-noise ratio (PSNR) and precision/recall [35] for inpaintings produced by our model. We find that Posterior Matching with VQ-VAE consistently achieves better precision/recall scores than previous models while having comparable PSNR.

**Hierarchical VAEs** Hierarchical VAEs [22, 37, 39] are a powerful extension of traditional VAEs that allow for more expressive priors and posteriors by partitioning the latent variables into subsets $\mathbf{z} = \{\mathbf{z}_1, \ldots, \mathbf{z}_L\}$. A hierarchy is then created by factorizing the prior $p(\mathbf{z}) = \prod_i p(\mathbf{z}_i \mid \mathbf{z}_{<i})$ and posterior $q(\mathbf{z} \mid \mathbf{x}) = \prod_i q(\mathbf{z}_i \mid \mathbf{z}_{<i}, \mathbf{x})$. These models have demonstrated impressive performance on images and can even outperform autoregressive models [8]. Posterior Matching can be

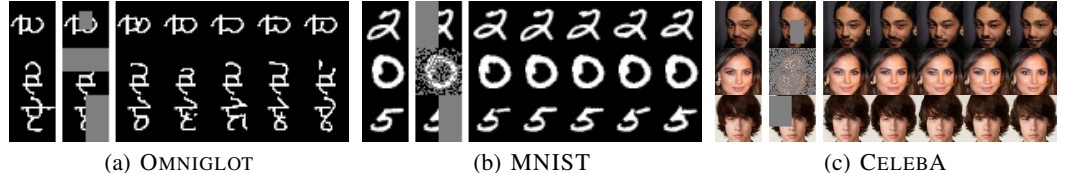

|  | (a) OMNIGLOT | (b) MNIST | (c) CELEBA |

Figure 5: Example image inpaintings from VDVAE with Posterior Matching. First two columns on the left are $\mathbf{x}$ and $\mathbf{x}_o$. Inpainting samples are on the right.

Table 2: Test normalized root-mean-square error (NRMSE) and arbitrary conditional log-likelihoods (LL) for UCI datasets. Lower is better for NRMSE, and higher is better for LL. Results for methods other than Posterior Matching are reproduced from Strauss and Oliva [38]. Mean and standard deviation are reported over 5 random observed masks for each instance.

| | POWER | | GAS | | HEPMASS | | MINIBOONE | | BSDS | |
| | NRMSE | LL | NRMSE | LL | NRMSE | LL | NRMSE | LL | NRMSE | LL |
|---|---|---|---|---|---|---|---|---|---|---|
| Posterior Matching | $0.834 \pm 0.001$ | $0.246 \pm 0.002$ | $0.330 \pm 0.013$ | $5.964 \pm 0.005$ | $0.857 \pm 0.000$ | $-8.963 \pm 0.007$ | $0.450 \pm 0.002$ | $-3.116 \pm 0.175$ | $0.573 \pm 0.000$ | $77.488 \pm 0.012$ |
| VAEAC | $0.880 \pm 0.001$ | $-0.042 \pm 0.002$ | $0.574 \pm 0.033$ | $2.418 \pm 0.006$ | $0.896 \pm 0.001$ | $-10.082 \pm 0.010$ | $0.462 \pm 0.002$ | $-3.452 \pm 0.067$ | $0.615 \pm 0.000$ | $74.850 \pm 0.005$ |
| ACE | $0.828 \pm 0.002$ | $0.631 \pm 0.002$ | $0.335 \pm 0.027$ | $9.643 \pm 0.005$ | $0.830 \pm 0.001$ | $-3.859 \pm 0.005$ | $0.432 \pm 0.003$ | $0.310 \pm 0.054$ | $0.525 \pm 0.000$ | $86.701 \pm 0.008$ |
| ACE Proposal | $0.828 \pm 0.002$ | $0.583 \pm 0.003$ | $0.312 \pm 0.033$ | $9.484 \pm 0.005$ | $0.832 \pm 0.001$ | $-4.417 \pm 0.005$ | $0.436 \pm 0.004$ | $-0.241 \pm 0.056$ | $0.535 \pm 0.000$ | $85.228 \pm 0.009$ |
| ACFlow | $0.877 \pm 0.001$ | $0.561 \pm 0.003$ | $0.567 \pm 0.050$ | $8.086 \pm 0.010$ | $0.909 \pm 0.000$ | $-8.197 \pm 0.008$ | $0.478 \pm 0.004$ | $-0.972 \pm 0.022$ | $0.603 \pm 0.000$ | $81.827 \pm 0.007$ |
| ACFlow+BG | $0.833 \pm 0.002$ | $0.528 \pm 0.003$ | $0.369 \pm 0.016$ | $7.593 \pm 0.011$ | $0.861 \pm 0.001$ | $-6.833 \pm 0.006$ | $0.442 \pm 0.001$ | $-1.098 \pm 0.032$ | $0.572 \pm 0.000$ | $81.399 \pm 0.008$ |

naturally applied to hierarchical VAEs, where the partially observed posterior is represented as $q(\mathbf{z} \mid \mathbf{x}_o) = \prod_i q(\mathbf{z}_i \mid \mathbf{z}_{<i}, \mathbf{x}_o)$. We adopt the Very Deep VAE (VDVAE) architecture used by Child [8] and extend it to include the partially observed posterior (see Appendix for training and architecture details). We note that due to our hardware constraints, we trained smaller models and for fewer iterations than Child [8]. Inpainting results for our VDVAE models are given in Table 1. We see that they achieve better precision/recall scores than the VQ-VAE models and, unlike VQ-VAE, are able to attain better PSNR than ACFlow for MNIST and CELEBA. Figure 5 shows some example inpaintings, and additional samples are provided in the Appendix. The fact that we see better downstream performance when using VDVAE than when using VQ-VAE is illustrative of Posterior Matching's ability to admit easy performance gains by simply switching to a more powerful base VAE.

### 5.3 Real-valued Datasets

We evaluate Posterior Matching on real-valued tabular data, specifically the benchmark UCI repository datasets from Papamakarios et al. [31]. We follow the experimental setup used by Li et al. [24] and Strauss and Oliva [38]. In these experiments, we train basic VAE models while simultaneously learning the partially observed posterior. Given the flexibility that Posterior Matching affords, we use an autoregressive distribution for $q_\theta(\mathbf{z} \mid \mathbf{x}_o)$. Further details can be found in the Appendix.

Table 2 reports the arbitrary conditional log-likelihoods and normalized root-mean-square error (NRMSE) of imputations for our models (with features missing completely at random). Likelihoods are computed using an importance sampling estimate (see Appendix for details). We primarily compare to VAEAC as a baseline in the VAE family, however we also provide results for ACE and ACFlow for reference. We see that Posterior Matching is able to consistently produce more accurate imputations and higher likelihoods than VAEAC. While our models don't match the likelihoods achieved by ACE and ACFlow, Posterior Matching is comparable to them for imputation NRMSE.

### 5.4 Partially Observed Clustering

Probabilistic clustering often views cluster assignments as a latent variable. Thus, when applying Posterior Matching in this setting, we may perform "partially observed" clustering, which clusters instances based on a subset of observed features. We consider VaDE, which uses a mixture of Gaussians as the prior, allowing it to do unsupervised clustering by treating each Gaussian component as one of the clusters [19]. Despite differences in how VaDE is trained compared to a classic VAE, training a partially observed encoder via Posterior Matching remains exactly the same.

We train models on both MNIST and FASHION MNIST (see Appendix for experimental details). Figure 6 shows the clustering accuracy of these models as the percentage of (randomly selected)

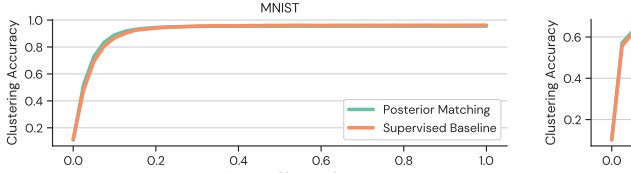
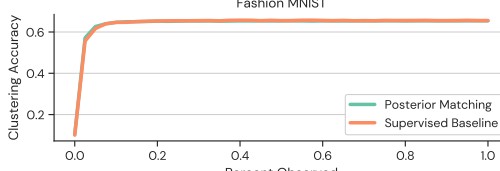

Figure 6: Partially observed clustering accuracy achieved by Posterior Matching with VaDE.

observed features changes. As a baseline, we train a supervised model where the labels are the cluster predictions from the VaDE model when all of the features are observed. We see that Posterior Matching is able to match the performance of the baseline, and even slightly outperform it for low percentages of observed features. Unlike the supervised approach, Posterior Matching has the advantage of being generative.

## 5.5 Very Fast Greedy Feature Acquisition

As discussed in Section 3.5, we can use Posterior Matching outside of the specific task of arbitrary conditioning. Here, we consider the problem of greedy active feature acquisition. We train a VAE with a Posterior Matching network that outputs the lookahead posteriors described in Section 3.5, using the loss in Equation 7. Note that we are also still using Posterior Matching in order to learn $q_\theta(\mathbf{z} \mid \mathbf{x}_o)$ and therefore to produce reconstructions. Training details can be found in the Appendix.

We consider the MNIST dataset and compare to EDDI as a baseline, using the authors' publicly available code. We downscale images to $16 \times 16$ since EDDI has difficulty scaling to high-dimensional data. We also only evaluate on the first 1000 instances of the MNIST test set, as the EDDI code was very slow when computing the greedy acquisition policy. EDDI also uses a particular architecture that is not compatible with convolutions. Thus we train a MLP-based VAE on flattened images in order to make a fair comparison. However, since Posterior Matching does not place any limitations on the type of VAE being used, we also train a convolutional version. For our models, we greedily select the

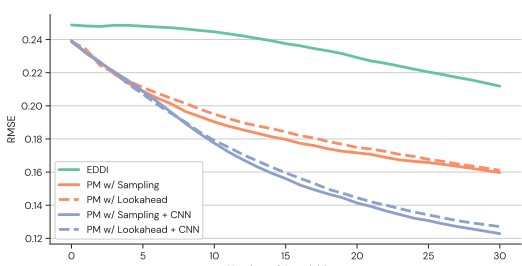

Figure 7: Average RMSE of reconstructions for greedy active feature acquisition.

feature to acquire using the more expensive sampling-based approach (similar to EDDI) as well as with the lookahead posteriors (which requires no sampling). In both cases, imputations are computed with an expectation over 50 latent codes, as is done for EDDI. An example acquisition trajectory is shown in Figure 3.

Figure 7 presents the root-mean-square error, averaged across the test instances, when imputing $\mathbf{x}_u$ with different numbers of acquired features. We see that our models are able to achieve lower error than EDDI. We also see that acquiring based on the lookahead posteriors incurs only a minimal increase in error compared to the sampling-based method, despite being far more efficient. Computing the greedy choice with our model using the sampling-based approach takes 68 ms $\pm$ 917 $\mu$s (for a single acquisition on CPU). Using the lookahead posteriors, the time is only 310 $\mu$s $\pm$ 15.3 $\mu$s, **a roughly 219x speedup**.

## 6 Conclusions

We have presented an elegant and general framework, called Posterior Matching, that allows VAEs to perform arbitrary conditioning. That is, we can take an existing VAE that only models the joint distribution $p(\mathbf{x})$ and train an additional model that, when combined with the VAE, is able to assess any likelihood $p(\mathbf{x}_u \mid \mathbf{x}_o)$ for arbitrary subsets of unobserved features $\mathbf{x}_u$ and observed features $\mathbf{x}_o$.

We applied this approach to a variety of VAEs for a multitude of different tasks. We found that Posterior Matching outperforms previous specialized VAEs for arbitrary conditioning with tabular data and for image inpainting. Importantly, we find that one can switch to a more powerful base VAE and get immediate improvements in downstream arbitrary conditioning performance "for free," without making changes to Posterior Matching itself. We can also use Posterior Matching to perform clustering based on partially observed inputs and to improve the efficiency of greedy active feature acquisition by several orders of magnitude at negligible cost to performance.

With this work, we hope to make arbitrary conditioning more widely accessible. Arbitrary conditioning no longer requires specialized methods, but can instead be achieved by applying one general framework to common VAEs. As advances are made in VAEs for joint density estimation, we can expect to immediately reap the rewards for arbitrary conditioning.

## Acknowledgments and Disclosure of Funding

We would like to thank Google's TPU Research Cloud program for providing free access to TPUs. This research was partly funded by NSF grant IIS2133595 and by NIH 1R01AA02687901A1.

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
