# OpenReview forum: "Posterior Matching for Arbitrary Conditioning"
_NeurIPS.cc/2022/Conference — NeurIPS 2022 Accept_

### Official Review · Reviewer_hezX · 2022-07-01

**Rating:** 6
**Confidence:** 5
**Soundness:** 3 good
**Presentation:** 3 good
**Contribution:** 3 good

**Summary:**

Authors propose a model and learning method, called posterior matching, to perform arbitrary conditioning with VAE models. The proposed method can be applied to a variety of VAE models without modifications to the VAE model itself, and can be used with a pre-trained VAE or can be trained alongside with the actual VAE model.

**Questions:**

List below contains both minor and major comments.

1. In Background section where you introduce the notation x_u and x_o, it would be very helpful to specifically mention that you assume fully observed datasets. A reader may easily get disoriented that you are dealing with partially observed data.

2. Equation. 1 in Section 3.2: From the point of view of the generative model, Eq. 1 is not exact but rather already an approximation of p(z|x_o) = p(x_o|z)p(z)/p(x_o) because your expression involves the variational approbation q(z|x).

3. Equation 2: it would be useful to more clearly define q_{\theta}, e.g. by referring to Fig. 2 already at this stage.

4. Equation 3: write already here that what are the parameters that you are optimizing.

5. At a general level, as summarised in Fig. 2, the proposed model contains two branches: the "original" VAE and the "proposed model part" that includes the partially observed encoder and provides the posterior matching loss. The amortized variational approximation of the "proposed model part" takes as an input the "observed" variables x_o (used for conditioning) as well as a binary vector b that indicates which variables are missing/non-missing. This proposed model part seems very similar with the models proposed to learn VAEs from partially observed data in (Collier et al, 2020; Ipsen et al, 2021). These methods can learn approximations of the posteriors p(z|x_o,b) and thereby used to infer p(x_u|x_o). Please contrast your model with these two e.g. in Section 4, and include them in ablation studies (see below).

REFS:
Mark Collier, Alfredo Nazabal, Christopher K.I. Williams. VAEs in the Presence of Missing Data pdf  Published on arXiv 13 July 2020. Presented at the first Workshop on the Art of Learning with Missing Values (Artemiss) hosted by the 37th International Conference on Machine Learning (ICML 2020).

NB Ipsen, PA Mattei, J Frellsen, not-MIWAE: Deep Generative Modelling with Missing not at Random Data, International Conference on Learning Representations, 2021.

6. Experiments and test metrics:
The general goal of the proposed method is to perform arbitrary conditioning p(x_u|x_o). I would have liked to see much more extensive experiments with variety of datasets that quantify performance directly using that metric, instead of devoting a large number of experiments to various other downstream tasks, where the contribution of the proposed model is less directly quantifiable. Current manuscript evaluate conditional log-likelihoods only in one experiment (Table 2).

Related to the comment on conditional log-likelihood evaluations: authors evaluate their method on various downstream tasks using a variety of evaluation metrics (peak SNR, precision, recall, clustering accuracy). While the metrics themselves are well-known, the way they are computed and applied here is not explained in detail, which leaves it unclear that what the numbers exactly tell about the proposed method itself.

7. Ablation studies:
I would have liked to see ablation studies. For example, it common to use the standard VAE or VAE with the Gaussian mixture prior and impute unobserved (missing) feature with zeros x_u = [0 0 ... 0], and then approximate the posterior directly using q(z|x_o,x_u), and then estimate p(x_u|x_o).  More recent methods explicitly model the missing features, such as the method by (Collier et al. 2020) and (Ipsen et al, 2021) (see comment above).  A comprehensive ablation study including e.g. these methods would provide more insight into the proposed model.

Overall, this is an interesting study that addresses an important problem. The proposed method is intuitive and statistically well-motivated, and results appear promising. Regarding the novelty, I would like to see the proposed method to be compared by recent works that propose closely related methods.  An ablation study and more straightforward, direct comparisons on the arbitrary conditioning would strengthen the manuscript.

**Limitations:**

Limitations are adequately addressed.

**Strengths And Weaknesses:**

Originality:
Previous work has proposed deep generative models for arbitrary conditioning using e.g. autoregressive and flow based models.  VAE based have also been proposed for arbitrary conditioning.  Regarding originality and novelty, this work seems to extend previous VAE methods by allowing more expressive posteriors as well as by allowing applying the proposed method to a pre-trained VAE model. Authors do not contrast their method to recent methods that allow learning VAEs from partially observed data (see detailed comments below).

Quality & Clarity:
Quality and clarity appears generally good. I have comments regarding the clarity of the experiments (see below).

Significance:
Significance of the work is very high. Arbitrary conditioning with (more classical) statistical models is of central importance, and the same applies to arbitrary conditioning with deep generative models. Consequently, this work is well motivated and has important applications.

---

> ### Author Response · Authors · 2022-08-02
> **Authors' Response**
>
> Thank you for your comments and suggested improvements. We think that clarifying with regards to your questions 1-4 will aid the readability of the paper, and we will happily update the text with the additional space allowed in the camera ready version.
>
> Thank you for pointing out two relevant references, which we will add discussion of to the camera ready version. The model introduced by Ispen et al. 2021 optimizes an approximation of $p(\mathbf{x}_o, \mathbf{b})$ and focuses on imputation for data that is missing “not at random”, a different setting that is outside the focus of our work, and their experiments therefore do not cover the same variety of tasks as our own. The workshop paper of Collier et al. 2020 similarly optimizes a lower bound on $p(\mathbf{x}_o \mid \mathbf{b})$. These approaches are in the same vein of the works we compared to such as, VAEAC and EDDI, and similarly they do not have the advantage of being compatible with pretrained “vanilla” VAEs.
>
> With regards to question 6, we note that we provide likelihood results on five datasets that represent a standard benchmark for generative models (see e.g. Masked Autoregressive Flow for Density Estimation NeurIPS 2017, Neural Autoregressive Flows ICML 2018). In the paper we also wanted to emphasize the fact that our method is able to be easily and successfully applied to a variety of downstream tasks. The metrics that we report for each of our downstream tasks are standard for those tasks: PSNR measures the average error in the image inpaintings, while precision and recall measure the inpaintings’ sample quality and coverage of the target distribution (Sajjadi et al. 2018). Clustering accuracy is a standard metric for unsupervised clustering models which tells us how well the model was able to group the data into the desired classes.
>
> For question 7, we note that the approach of simply imputing zeros for missing elements and then using the original base VAE suffers greatly from distribution shift, because the VAE was never trained to encounter those types of inputs. Thus, it is expected that this approach typically has worse performance. We have updated the appendix to include some preliminary experiments illustrating this, where we see poor imputation quality. Please see appendix E in the now updated supplementary material if you would like to view these results.

---

### Official Review · Reviewer_5R6L · 2022-07-04

**Rating:** 8
**Confidence:** 3
**Soundness:** 4 excellent
**Presentation:** 4 excellent
**Contribution:** 3 good

**Summary:**

This paper presents a general approach for adding arbitrary conditioning to models based on the VAE architecture. An additional encoder which takes partially observed inputs is used to define a partially observed posterior which is trained to match the fully observed posterior by adding an additional term to the loss function. The simplicity of this approach allows one to train a partially observed posterior for any VAE-based model and even to learn a partially observed posterior for an existing trained VAE.

The authors demonstrate the effectiveness of their approach on a number of tasks, using MNIST to illustrate the model matches their motivating intuitions and then using image inpainting, imputation of tabular data, and partially observed clustering to demonstrate the effectiveness of their method on tasks involving arbitrary conditioning. The method presented in this paper does not achieve top performance on all tasks, but is competitive with the SOTA and improves on other VAE-based methods. Other methods are less flexible, in the sense that they cannot be immediately integrated with a new VAE architecture, one clear appeal of the method presented here.

Additionally, the authors apply their method can be used for greedy active feature acquisition, where a conditional distribution based on partially observed features is used to greedily select the most informative feature to add to the model. Here, the method is shown to be more accurate and faster than existing methods.

**Questions:**

No questions to the authors

**Limitations:**

One limitation, which the authors remark on, is that this method requires fully observed training data. While there may be cases where it is advantageous to be able to train a model on partially observed data, there are many cases where some fully observed data will be available at training time.

**Strengths And Weaknesses:**

The paper is based on a simple but novel idea, which is clearly explained and validated using well thought-out experiments.

---

> ### Author Response · Authors · 2022-08-02
> **Authors' Response**
>
> We appreciate you reviewing our work and are glad you found it useful and general.

---

### Official Review · Reviewer_HjaN · 2022-07-09

**Rating:** 6
**Confidence:** 4
**Soundness:** 4 excellent
**Presentation:** 4 excellent
**Contribution:** 2 fair

**Summary:**

This manuscript proposes a method "Posterior Matching" that trains Variational Autoencoders (VAEs) to be robust to missing data when they are trained with fully observed data.  The idea is straightforward to implement and include with a wide variety of VAEs, and leads to relatively straightforward algorithms.  Theory demonstrates that the method attempts to approximate arbitrary conditional distributions.  Limiting to a subset of VAE approaches, the posterior matching method demonstrates improved performance on a number of benchmarks.

**Questions:**

How feasible is it to extend this approach to a case where the data is not fully observed?

Can the in-painting examples in fact capture the uncertainty in digits as motivating in Figure 1?

How well can this method handle missingness that isn't missing completely at random?

What is the optimal strategy for picking the missingness patterns during training?  How would this be determined for a new dataset?

**Limitations:**

The authors clearly state that they assume that they have access to fully observed data for training, and they correctly identify that as a limitation; however, they state that "this is generally not a relevant limitation for our purposes." This limitation is quite large and is downplayed too much in their manuscript.  For example, in the motivating medical case given in the introduction, this would render the method infeasible.  I work with medical data, and I am unaware of any problem within that domain where I could naturally apply their methods, whereas I can and do apply competing missing data approaches for VAEs.  I would suggest that the authors clarify this limitation upfront in the introduction, at least, and discuss it at greater length rather than making it an small comment in a technical section.

The other major limitation, which was not addressed, is that this method assumes a "Missing Completely at Random" model of missingness.  This model is incorrect for many medical problems, and would be an issue on the motivating example in the introduction.  I do not see how to adapt the Posterior Matching approach to handle this case, but it would be helpful for the authors to clearly state this limitation to give the readers a better sense of when this method will and will not work.

**Strengths And Weaknesses:**

The biggest strength of this paper is that it is a theoretically motivated algorithm that is simple to include in many VAE models, and can especially work with more complex variants such as VADE.  While there are two glaring limitations (see next section), this is a big advantage and the empirical results suggest that the method is effective.  The simplicity of the active learning approach is quite nice as well.

A weakness is the relative lack of comparisons.  While Posterior Matching is compared to multiple algorithms within the VAE space, there are many competing approaches for applications such as inpainting.  It seems clear that the proposed approach can improve in this cases, but it is unclear if I should consider their approach for an inpainting task.

There are several minor issues.  For example, the notation is a bit inconsistent.  As an example, in equation (1) the partially observed posterior is written in terms of the variational posterior; that is explicitly true as the variational distribution is an approximation and they are not exactly equal.  This seems like the authors meant that this term should be similar to $q_\theta(z|x_0)$ instead.  Either way, this should be clarified and explained more.

As a comment, given the motivating case in Figure 1, it would have been nice to see the proposed approach maintain the multimodality in the generated examples of Figure 5(b).  I would appreciate getting stronger evidence that the multi-modality posterior is in fact maintained.

---

> ### Author Response · Authors · 2022-08-02
> **Authors' Response**
>
> Thank you for your feedback. We address your questions below:
> - In our setting, we considered having access to a base “vanilla” VAE that was trained traditionally (i.e. with fully observed data). If one were not able to train the base VAE with fully observed data, then one approach would be to instead train the base VAE using an approach that bounds the marginals $p(\mathbf{x}_o)$, for observed features $\mathbf{x}_o$ (e.g., EDDI). After, posterior matching can be done taking the sets $u$ and $o$ from available features during training.
> - Yes, we find that Posterior Matching captures the multimodality of the unobserved features. We actually studied the types of uncertainty in Figure 1 using our model in Figure 4. For example, see the second column from the left of Figure 4, where the model produces inpaintings that span numerous different digits. We see that for an observation of small line portion (top of second column) codes from 1’s, 4’s, 7’s, and 9’s are inferred); similarly, for an observation of a curved portion (bottom of second column) codes from 2’s, 3’s, 8’s, and even odd slanted 1’s are inferred.
> - We will clarify our assumption of missing completely at random, which is similar to that of previous work such as VAEAC, ACFlow, and ACE, in the final version. Modeling more complicated missingness would require extending our work and incorporating other techniques (e.g. not-MIWAE).
> - One may draw missingness patterns according to the anticipated distribution of patterns that will be seen during deployment. However, in lieu of any anticipated distribution we found that sampling masks uniformly at random works well in general. In images, previous works (e.g. VAEAC, ACFlow) have noted that further incorporating contiguous masks is beneficial.
>
> We have also updated the notation and text around Equation 1 to more clearly convey our intended meaning.

---

> > ### Comment · Reviewer_HjaN · 2022-08-08
> > **Thanks for the response.**
> >
> > I am largely satisfied with this response.  I appreciate the additional clarity on limitations in the manuscript.
> >
> > Figure 4, second column is nice. I would suggest highlighting the multi-modality more in the draft, as I find it easy to miss.

---

### Official Review · Reviewer_cCKE · 2022-07-10

**Rating:** 7
**Confidence:** 3
**Soundness:** 4 excellent
**Presentation:** 4 excellent
**Contribution:** 3 good

**Summary:**

This work proposes a simple method for the arbitrary conditioning problem, which trains an amortized inference network for the distribution $q(z | x_o)$. Compared with previous work, the method does not require reparameterizable inference networks, and thus enables the use of more flexible models.

**Post-rebuttal update**: I am satisfied by the authors' response, and believe this work is suitable for publication.

**Questions:**

I do not have any major questions.  A few minor questions / comments:

- You should cite the early works on data imputation with VAEs, e.g. Rezende, Mohamed & Wierstra (2014, Appendix F).

- In Theorem 3.1 the definitions of $p_{\theta,\phi}, q_\theta(z | x_o, x_u)$ should be moved to the text.  The latter distribution should also depend on $\phi$ (the definition of $q_\theta$ does not involve $x_u$).  As both terms depend on $\theta$, ideally there should be more discussion about the optima, as mentioned in the above section.

**Limitations:**

Limitations and potential negative societal impact are adequately addressed.

**Strengths And Weaknesses:**

**Strengths**:

The method is simple, intuitive and appears to have competitive performance.

**Weaknesses**:

I do not have any major concerns.  My only gripe is that Theorem 3.1 appears to add little value -- the distributions to be optimized depend on $\theta$ in a somewhat complicated way, and the only thing I can read out of Eq. (4) is that when both $p_\phi$ and $q_\psi$ are optimized, the imputation distribution determined by $q_\theta$ will be correct; but this already follows from Eq. (3).  More discussions on the general case would be helpful.

---

> ### Author Response · Authors · 2022-08-02
> **Authors' Response**
>
> Thank you for the pointer to Rezende, Mohamed & Wierstra 2014. We will happily refer to this work in the camera ready version with the additional allotted space. Our intent with Theorem 3.1 is to show that Posterior Matching optimizes the arbitrary conditioning likelihood $p(\mathbf{x}_u \mid \mathbf{x}_o)$. While Posterior Matching is intuitive, the mathematical connection with arbitrary conditioning is implicit without Theorem 3.1.

---

### Author Response · Authors · 2022-08-02
**Authors' Rebuttal**

We would like to thank all the reviewers for their time and informative comments. We hope that our paper is recommended for publication as we believe that this work represents a novel contribution in reasoning about latent variables flexibly with varying conditioning covariates, which would be of interest to the ML community.

Please see reviewer specific comments posted below.

---

### Meta-Review · Area_Chair_JvK4 · 2022-08-25

**Recommendation:** Accept
**Confidence:** Less certain

**Metareview:**

The paper presents a method for conditional generation of part of the data variable given the rest part, where the joint data distribution is defined by a trained VAE. The partition can be made arbitrary. A model standing alone from the VAE is trained. Extension for faster active feature acquisition is also presented. All reviewers acknowledged the significance of the task and the simplicity/flexibility and empirical effectiveness of the method. Notably multi-modality generation results are seen. Reviewers also expressed concerns on the notation, clarity (e.g., insufficient emphasis on requiring fully observed data), and conceptual and empirical comparison with similar methods. The authors have addressed most of them. In all, this paper makes an interesting contribution to the community.

The authors are encouraged to include the discussions with reviewers into the paper (particularly, relation and comparison with similar methods mentioned). It would make Theorem 3.1 more insightful if the authors could explain how the second term in Eq. (4) makes the method better than directly optimizing the arbitrary conditioning likelihood. Moreover, it seems the method requires $x_u$ and $x_o$ are conditionally independent given $z$ (so that $p(x_u | x_o, z) = p(x_u | z)$). Though Line 163 mentioned factorization, it is not for addressing this point.

**Award:**

No

---

### Decision · Program_Chairs · 2022-09-14

Accept